# MEKK2 mediates aberrant ERK activation in neurofibromatosis type I

Seoyeon Bok [1,8], Dong Yeon Shin[1,2,8], Alisha R. Yallowitz[1], Mark Eiseman[1], Michelle Cung [1], Ren Xu[3], Na Li[3], Jun Sun [1], Alfred L. Williams [4], John E. Scott [4], Bing Su[5,6], Jae-Hyuck Shim [7] & Matthew B. Greenblatt [1✉]

Neurofibromatosis type I (NF1) is characterized by prominent skeletal manifestations caused by *NF1* loss. While inhibitors of the ERK activating kinases MEK1/2 are promising as a means to treat NF1, the broad blockade of the ERK pathway produced by this strategy is potentially associated with therapy limiting toxicities. Here, we have sought targets offering a more narrow inhibition of ERK activation downstream of NF1 loss in the skeleton, finding that MEKK2 is a novel component of a noncanonical ERK pathway in osteoblasts that mediates aberrant ERK activation after NF1 loss. Accordingly, despite mice with conditional deletion of *Nf1* in mature osteoblasts (*Nf1^{fl/fl};Dmp1-Cre*) and *Mekk2^{−/−}* each displaying skeletal defects, *Nf1^{fl/fl};Mekk2^{−/−};Dmp1-Cre* mice show an amelioration of NF1-associated phenotypes. We also provide proof-of-principle that FDA-approved inhibitors with activity against MEKK2 can ameliorate NF1 skeletal pathology. Thus, MEKK2 functions as a MAP3K in the ERK pathway in osteoblasts, offering a potential new therapeutic strategy for the treatment of NF1.

[1] Department of Pathology and Laboratory Medicine, Weill Cornell Medicine, New York, NY 10065, USA. [2] Research Center, LegoChem BioSciences, INC., Daejeon 34302, South Korea. [3] State Key Laboratory of Cellular Stress Biology, School of Medicine, Xiamen University, 361102 Xiamen, Fujian, China. [4] Department of Pharmaceutical Sciences, Biomanufacturing Research Institute and Technology Enterprise (BRITE), North Carolina Central University, Durham, NC 27707, USA. [5] Department of Immunology and Microbiology, and the Minister of Education Key Laboratory of Cell Death and Differentiation, Shanghai Institute of Immunology, Shanghai JiaoTong University School of Medicine, 200025 Shanghai, China. [6] Yale Institute for Immune Metabolism, Shanghai JiaoTong University School of Medicine, 200025 Shanghai, China. [7] Division of Rheumatology, Department of Medicine, University of Massachusetts Medical School, 364 Plantation Street, Worcester, MA 01605, USA. [8] These authors contributed equally: Seoyeon Bok, Dong Yeon Shin. ✉email: mag3003@med.cornell.edu

Neurofibromatosis type I (NF1) is an autosomal dominant disorder with an incidence of 1/3500, and ~50% of patients with NF1 display skeletal manifestations including craniofacial dysmorphogenesis, osteopenia/osteoporosis, and impaired fracture healing, including congenital pseudoarthroses. The *NF1* gene product, neurofibromin, is a Ras GTPase-activating protein that functions as a negative regulator of Ras/ERK signaling in the development and growth of a variety of tissues[1]. Loss-of-function mutations in *NF1* result in hyperactivation of several signaling pathways, prominently including the MEK/ERK pathway, and preclinical studies indicate utility for MEK/ERK pathway inhibition for treating skeletal and other organ manifestations of NF1[2–4]. Along these lines, ERK pathway inhibitors have recently been shown to have clinical utility for the treatment of a subset of plexiform neurofibromas in NF1[5]. Despite this promise, studies of ERK pathway inhibition in other clinical contexts suggest that it can be associated with serious and likely therapy limiting toxicities in a subset of patients, prominently cardiovascular complications including myocardial infarction, heart failure, cardiomyopathy, and hypertension[6–8]. These toxicities are likely intrinsic to the broad importance of the ERK/MAPK pathway in controlling fundamental cellular processes such as proliferation and survival. Toxicity concerns are exacerbated when considering that therapy for skeletal manifestations of NF1 would most plausibly entail long-term or even lifelong treatment. Moreover, profound blockade of ERK pathway activity is associated with severe osteopenia, raising concerns that non-selective ERK pathway inhibition could actually exacerbate skeletal pathology in NF1, or at least that ERK pathway inhibition is expected to have a narrow therapeutic window with regards to skeletal endpoints[9–11]. To overcome these limitations, more selective approaches are needed to target pathways mediating the downstream effects of NF1 loss-of-function.

MAPKs operate in a three-tiered cascade: MAP kinase kinase kinases (MAP3K), MAP kinase kinases (MAP2K), and MAPKs. While the connections between MAP2Ks and MAPKs largely occur in a stereotypical and invariant manner, the connections between MAP3Ks and downstream MAP2Ks are much more context-dependent, differing both among cell types and with respect to specific stimuli. This suggests that inhibition of MAP3Ks may offer a more selective and ultimately less toxic means than directly targeting MAP2Ks or MAPKs to inhibit the aberrant ERK pathway activation occurring downstream of NF1 loss-of-function. While MAP3Ks contributing to p38 and JNK activation in osteoblasts, including ASK1 (MAP3K5)[12] and TAK1 (MAP3K7)[13,14], have been identified, the MAP3K mediating ERK activation in osteoblasts is largely unknown, aside from studies showing that MLK3 (MAP3K11) contributes to both ERK and p38 activation and a study of RAF isoforms largely focused on cartilage[15,16]. MEKK2 (MAP3K2) is a member of the MEK kinase group of MAP3Ks, and early in vitro studies demonstrated that MEKK2 has the ability to activate a number of downstream MAPK pathways, including ERK1/2, JNK, p38, and ERK5[17–21]. We previously observed that MEKK2 mediated an alternative pathway for the deubiquitination and stabilization of β-catenin in osteoblasts and that MEKK2-deficient mice display significant cortical and trabecular osteopenia due to impaired osteoblast activity[22].

Here we identify MEKK2 as a MAP3K contributing to ERK pathway activation in osteoblasts and show that loss of MEKK2 is sufficient to prevent the constitutive ERK activation occuring in models of skeletal NF1. Accordingly, loss of MEKK2 can ameliorate the skeletal manifestations occuring in a mouse model of skeletal NF1. We also show that ponatinib, an FDA-approved tyrosine-kinase inhibitor, can inhibit MEKK2 and that administration of ponatinib significantly improves skeletal pathology in a mouse model of skeletal NF1. This provides proof-of-principle for targeting MEKK2 as a strategy for the management of the skeletal manifestations of NF1.

## Results

**MEKK2 mediates ERK activation downstream of NF1 loss-of-function.** Given the ability of MEKK2 to activate the ERK pathway in previous overexpression studies conducted in a non-skeletal context[17,21], we considered it as a candidate MAP3K mediating ERK activation in osteoblasts. Immunoblotting showed that ERK1/2 activation was markedly reduced in primary $Mekk2^{-/-}$ calvarial osteoblasts (COBs), and, accordingly, that phosphorylation of the well-characterized ERK substrates RSK2 and GSK3β was also reduced (Fig. 1a). RSK2 is known to in turn phosphorylate and activate ATF4, a critical transcription factor regulating the later stages of osteoblast differentiation[23]. Consistent with this reduction in RSK2 activation, $Mekk2^{-/-}$ COBs showed reduced activity of the ATF4-responsive osteoblast-specific cis-acting element 1 (OSE1)-luciferase reporter plasmid (Fig. 1b). To provide direct biochemical confirmation of the ability of MEKK2 to mediate ERK pathway activation, an in vitro kinase assay was performed, showing that recombinant MEKK2 directly phosphorylated MEK1 and MEK2 (Fig. 1c). Thus, MEKK2 contributes to basal ERK activation, and NF1 loss-of-function via lentiviral Cre deletion in $Nf1^{fl/fl}$ osteoblasts induces aberrant ERK activation (Fig. 1d). To examine aberrant ERK activation downstream of NF1 loss-of-function in vivo, we generated mice with a conditional deletion of *Nf1* in mature osteoblasts using dentin matrix acidic phosphoprotein 1-Cre (*Dmp1-Cre*)[24]. Consistent with in vitro results, ERK was highly activated in osteoblasts in the femurs of $Nf1^{fl/fl}$;*Dmp1-Cre* mice compared to $Nf1^{fl/fl}$ controls (Fig. 1e). We next examined if MEKK2 mediates ERK activation downstream of NF1 loss-of-function. Previously, we observed that FGF2 strongly activates MEKK2 phosphorylation in osteoblasts[22]. Indeed, knockdown of *NF1* in human mesenchymal stromal cell-derived osteoblasts resulted in enhanced MEKK2 phosphorylation in response to FGF2, showing that loss of NF1 can enhance MEKK2 activation (Fig. 1f). Similarly, transduction of $Nf1^{fl/fl}$ osteoblasts with lentiviral Cre increased MEKK2 phosphorylation as shown by either use of a phos-tag gel or immunoblotting with anti-p-MEKK2 antibodies (Fig. 1g). Immunohistochemistry for p-MEKK2 levels in cortical bone was performed, finding that $Nf1^{fl/fl}$;*Dmp1-Cre* mice showed greater positivity for p-MEKK2-within bone adjacent mesenchymal cells (Fig. 1h). Likewise, knockdown of *Nf1* enhanced ERK activation by FGF2, and this enhancement was rescued by additional knockdown of *Mekk2* (Fig. 1i). Thus, MEKK2 contributes to the aberrant hyperactivation of ERK occurring with NF1 loss-of-function in vitro.

**MEKK2 deficiency rescues NF1-associated skeletal phenotypes.** Previously, $Nf1^{fl/fl}$;*Dmp1-Cre* mice have been reported as a model of skeletal NF1, finding that they display spontaneous fractures, accompanied by reduced mechanical strength, low bone mineral density, and high cortical porosity with an osteomalacia-like bone phenotype[25]. Use of the *Dmp1-Cre* here has the advantage that it avoids the severe growth and joint dysplasia phenotypes associated with the deletion of *NF1* in early osteoprogenitors/skeletal stem cells as seen with *Prx1* or *Col2-Cre*, which may complicate interpretation of genetic or pharmacologic rescue experiments[26,27]. In line with prior reports, we observed that $Nf1^{fl/fl}$;*Dmp1-Cre* mice display high cortical porosity and uneven deposition of endosteal cortical bone. Based on our in vitro findings, we hypothesized that additional deficiency for MEKK2 will ameliorate these phenotypes in $Nf1^{fl/fl}$;*Dmp1-Cre* mice.

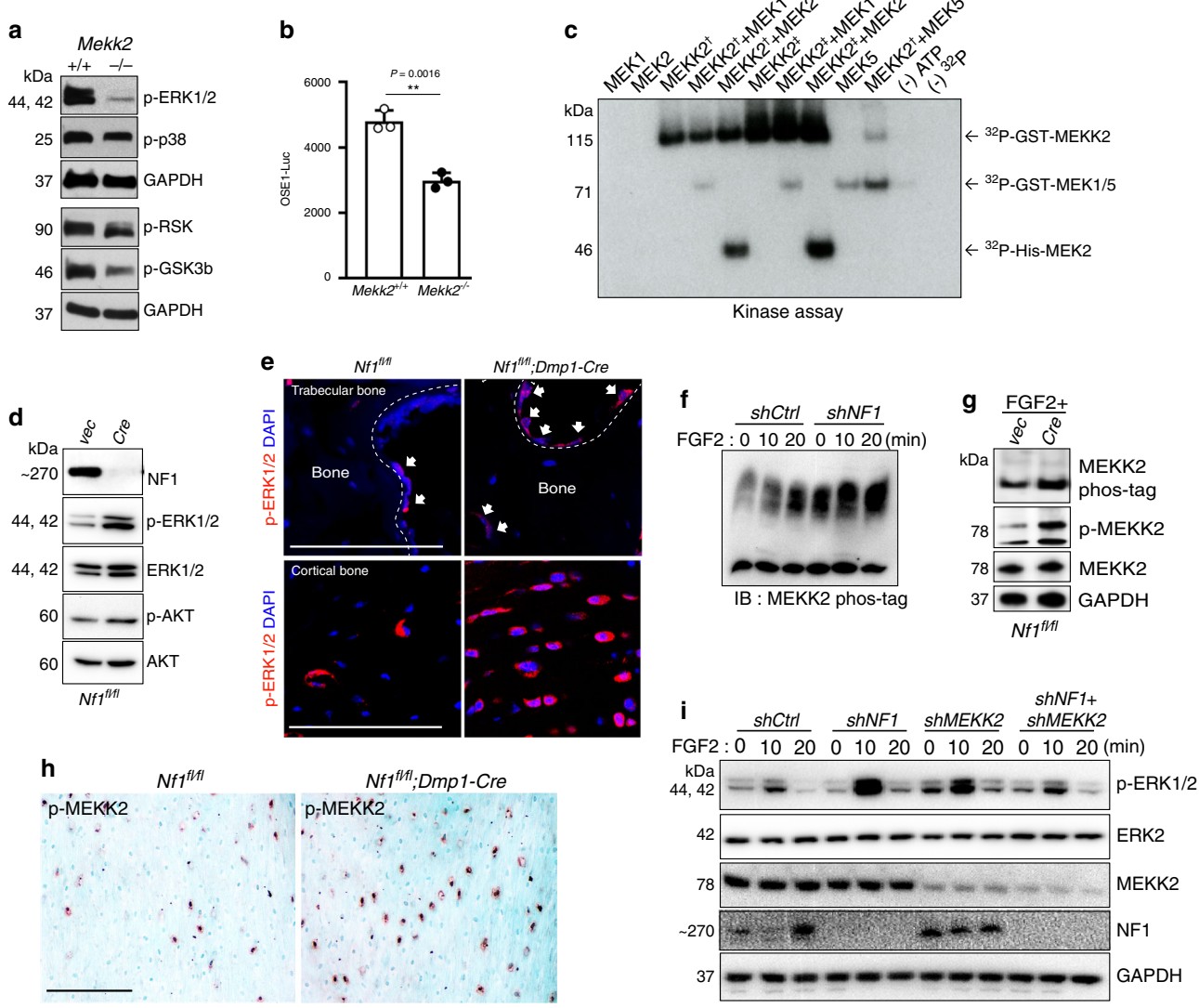

**Fig. 1 The MAP3K MEKK2 contributes to the aberrant ERK activation occurring with NF1 loss. a** ERK phosphorylation levels were assessed by immunoblotting in primary COBs isolated from *Mekk2+/+* and *Mekk2−/−* mice. Cell lysates were immunoblotted with the indicated antibodies. **b** Primary *Mekk2+/+* and *Mekk2−/−* COBs were transfected with the OSE1-Luc (ATF4) reporter (*n* = 3 biologically independent samples). Mean ± s.e.m., unpaired, two-tailed Student's *t* test: \*\**P* < 0.01. **c** Purified unactivated GST-MEK1 or His-MEK2 was incubated with purified GST-MEKK2 and an in vitro kinase assay was conducted. Two types of recombinant GST-tagged MEKK2 as indicated with a dagger and double dagger were used as described in the Methods section. GST-MEK5 was used as a positive control. **d** Representative blot from three independent experiments. Primary *Nf1fl/fl* COBs infected with either vector (Vec) or Cre lentivirus, were cultured for 7 days under differentiation conditions. Levels of NF1, phospho-ERK1/2, and ERK1/2 were analyzed by immunoblotting. **e** Representative images of immunostaining for p-ERK (red) in femurs from 16 weeks old *Nf1fl/fl* and *Nf1fl/fl;Dmp1-Cre* mice. White arrows indicate p-ERK positive osteoblasts. Nuclei are counterstained with DAPI (blue) and scale bar indicates 100 μm. Three independent fields were examined per mouse (*n* = 3 mice per group). **f** human MSCs (hMSCs) were infected with shRNA lentiviruses expressing control (shCtrl) or *NF1* (shNF1) targeting shRNAs and stimulated with FGF2 (25 ng/ml) for the indicated times, then MEKK2 phosphorylation was analyzed by phos-tag electrophoresis. **g** MEKK2 phosphorylation was analyzed by either phos-tag electrophoresis or immunoblotting with p-MEKK2 antibody in primary *Nf1fl/fl* osteoblasts infected with either vec or Cre lentivirus cultured for 14 days under differentiation conditions. All blotting was confirmed by at least three independent repeats. **h** Immunohistochemistry for p-MEKK2 was performed in femurs from 16-week-old *Nf1fl/fl* and *Nf1fl/fl;Dmp1-Cre* mice. Scale bar denotes 100 μm. Three independent fields were examined per mouse (*n* = 3 mice per group). **i** hMSCs were infected with the indicated shRNA lentiviruses and then stimulated with FGF2 for the indicated times before immunoblotting, and ERK1/2 activation was analyzed by immunoblotting. Except where otherwise indicated, all data shown are representative of at least two independent experiments. All unprocessed blots are provided in Supplementary Fig. 5. Source data are provided as a Source Data file.

Indeed, despite *Mekk2−/−* mice themselves being osteopenic and having impaired bone formation, *Nf1fl/fl;Mekk2−/−;Dmp1-Cre* mice displayed a substantial rescue of the *Nf1fl/fl;Dmp1-Cre* phenotype, with normalization of cortical architecture and porosity (Fig. 2a, b), trabecular osteopenia (Fig. 2c, d) and calvarial hypomineralization (Fig. 2e). In keeping with this, levels of P1NP (procollagen type 1 N-terminal propeptide), a marker for

bone formation and osteoblast activity, were greatly increased in *Nf1fl/fl;Mekk2−/−;Dmp1-Cre* mice compared to *Nf1fl/fl;Dmp1-Cre* mice, whereas no significant difference was observed in levels of CTX (Collagen type I C-terminal telopeptide), a marker of osteoclast activity[28]. Similarly, FGF23, 25(OH) vitamin D, and phosphate levels were all normalized in *Nf1fl/fl;Mekk2−/−;Dmp1-Cre* mice relative to *Nf1fl/fl;Dmp1-Cre* mice (Fig. 2f). Consistent

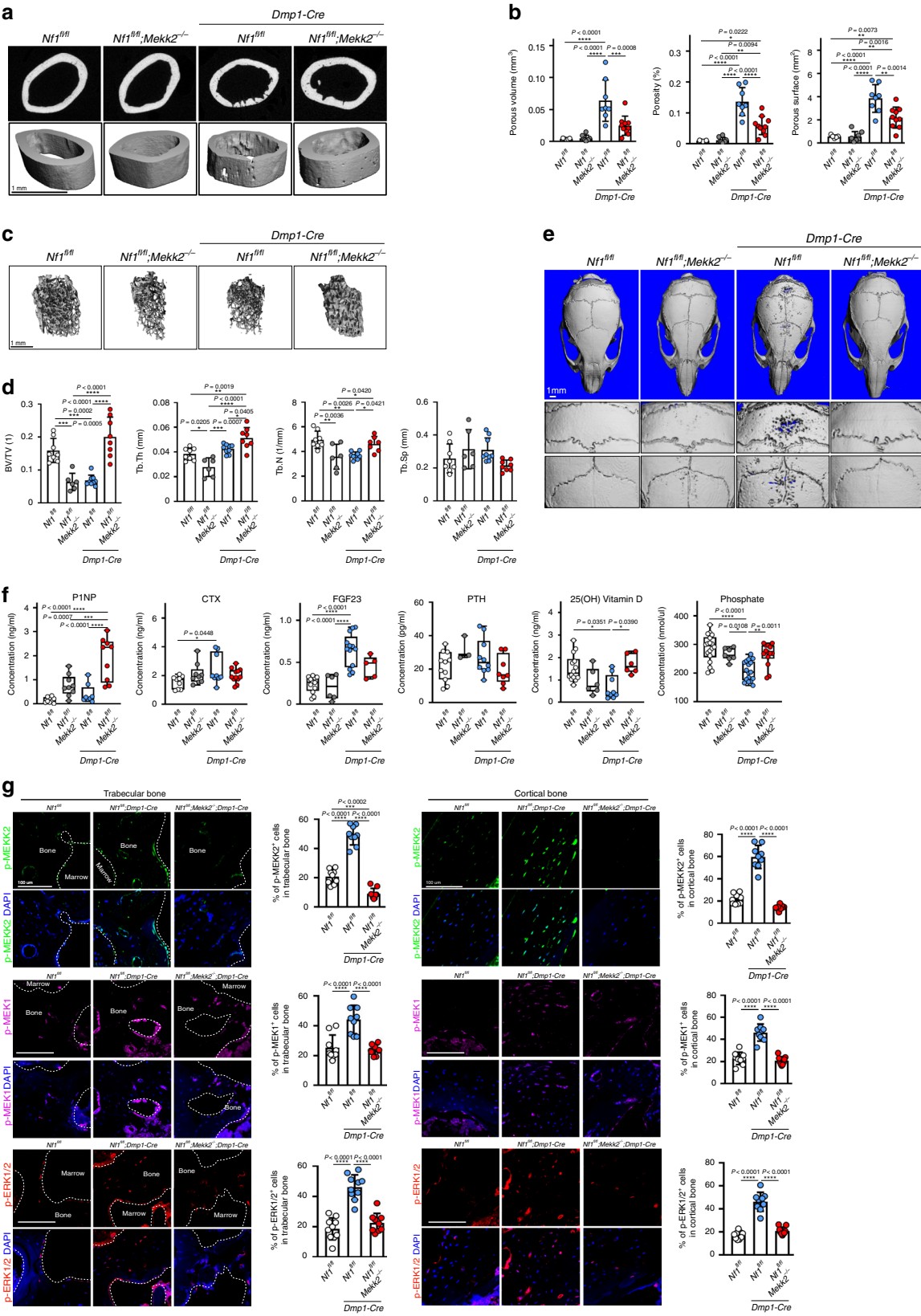

with in vitro observations of MEKK2 acting downstream of NF1 loss-of-function to mediate ERK pathway activation, immunostaining confirmed that p-MEKK2, p-MEK1, and p-ERK1/2 levels were strongly decreased in osteoblasts in femurs of *Nf1*[fl/fl]*;Mekk2*[−/−]*;Dmp1-Cre* mice compared to *Nf1*[fl/fl]*;Dmp1-Cre* mice (Fig. 2g). p-AKT levels were also analyzed by

immunostaining, but there were no significant differences among these mice (Supplementary Fig. 1a). To explore impacts on the osteocyte dendritic network, we performed filamentous actin (F-actin) staining, finding clear disruption of the osteocyte dendritic network in *Nf1*[fl/fl]*;Dmp1-Cre* mice. However, this aspect of the NF1 loss-of-function phenotype did not show a substantial degree

**Fig. 2 MEKK2 deficiency partially reverses NF1-associated skeletal pathology in mice. a** Femurs from 16-week-old $Nf1^{fl/fl}$, $Nf1^{fl/fl};Mekk2^{-/-}$, $Nf1^{fl/fl};Dmp1$-$Cre$, and $Nf1^{fl/fl};Mekk2^{-/-};Dmp1$-$Cre$ mice were analyzed by μCT. **b** Quantitative parameters are porous volume, porosity (porous volume/total volume), and porous surface from 16-week-old $Nf1^{fl/fl}$ ($n = 5$), $Nf1^{fl/fl};Mekk2^{-/-}$ ($n = 8$), $Nf1^{fl/fl};Dmp1$-$Cre$ ($n = 8$), and $Nf1^{fl/fl};Mekk2^{-/-};Dmp1$-$Cre$ ($n = 10$) mice. Mean ± s.d., one-way ANOVA with Tukey's multiple comparison test. **c** Representative μCT 3D-reconstuction images of trabecular bone in the distal femur metaphysis. **d** Quantitative parameters include trabecular bone volume/total volume (BV/TV), thickness (Tb.Th), trabecular number (Tb.N), and spacing (Tb.sp) in 16-week-old $Nf1^{fl/fl}$ ($n = 8$), $Nf1^{fl/fl};Mekk2^{-/-}$ ($n = 6$), $Nf1^{fl/fl};Dmp1$-$Cre$ ($n = 9$), and $Nf1^{fl/fl};Mekk2^{-/-};Dmp1$-$Cre$ ($n = 8$) mice. mean ± s.d., one-way ANOVA with Tukey's multiple comparison test. **e** μCT scans of mouse skulls at 16 weeks $Nf1^{fl/fl}$, $Nf1^{fl/fl};Mekk2^{-/-}$, $Nf1^{fl/fl};Dmp1$-$Cre$, and $Nf1^{fl/fl};Mekk2^{-/-};Dmp1$-$Cre$ mice. **f** Serum levels of P1NP, CTX, FGF23, PTH, 25(OH) vitamin D, and phosphate in 16-week-old mice. Data are represented as box plots with the middle line representing the median, the box representing the 95% confidence interval of the median, and the whiskers representing the range. Each dot represents a separate mouse. **g** Representative images and quantification of immunostaining for p-MEKK2 (green), p-MEK1 (magenta), and p-ERK (red) in trabecular (left panels) and cortical bone (right panels) from 16-week-old $Nf1^{fl/fl}$, $Nf1^{fl/fl};Dmp1$-$Cre$, and $Nf1^{fl/fl};Mekk2^{-/-};Dmp1$-$Cre$ mice. Nuclei are counterstained with DAPI (blue) and the scale bar indicates 100 μm. Three independent fields were examined per mouse ($n = 3$ mice per group). mean ± s.d., one-way ANOVA with Tukey's multiple comparison test. $*P < 0.05$; $**P < 0.01$; $***P < 0.001$; $****P < 0.0001$.

of rescue by additional MEKK2 deficiency (Supplementary Fig. 1b). Thus, deficiency for MEKK2 ameliorates a wide range of NF1-associated skeletal phenotypes in mice.

**Ponatinib inhibits MEKK2 and ameliorates NF1 phenotypes.** Given this rescue, we hypothesized that pharmacological inhibition of MEKK2 could have clinical utility in treating the skeletal manifestations of NF1. To maximize the translational relevance of this study, we focused on identifying whether well-studied and FDA-approved kinase inhibitors had activity against MEKK2, focusing on a set of compounds previously studied in cell-free assays with MEKK2, including AT9283, crizotinib, ponatinib, and bosutinib[29,30]. By immunoblotting, we observed that ponatinib strongly reduced MEKK2 phosphorylation (Fig. 3a–c) and further observed that ERK activation was completely inhibited by AT9283 and ponatinib in response to FGF2 (Fig. 3d, e). An in vitro kinase assay using purified recombinant proteins confirmed that ponatinib inhibited both MEKK2 autophosphorylation and MEKK2-mediated MEK1 phosphorylation in a dose-dependent manner with an IC$_{50}$ of ~100 nM (Fig. 3f). To examine whether ponatinib phenocopies the effects of $Mekk2$ knockdown, mRNA levels were analyzed in NF1-deficient osteoblasts during in vitro differentiation. Osteoblast-related genes, including $Osx$, $Runx2$, $Bsp$, and $Ocn$ were measured, demonstrating that either Mekk2 knockdown or ponatinib treatment both rescued or increased expression of osteoblast marker genes (Fig. 3g).

To strengthen the evidence that these in vitro effects of ponatinib to block MEKK2-mediated ERK pathway activation downstream of NF1 loss-of-function truly reflect inhibition of MEKK2, a compound with a distinct chemical scaffold expressly identified as a MEKK2 inhibitor was examined (BRITE-719)[31]. BRITE-719 had similar effects to those of ponatinib, strongly inhibiting ERK phosphorylation (Fig. 3h). BRITE-690, an analog of BRITE-719 lacking MEKK2 inhibitory activity, showed no impact in these assays, further confirming that the effects observed reflect inhibition of MEKK2.

After validating that ponatinib has the activity to inhibit MEKK2, we next examined whether ponatinib can rescue the skeletal phenotypes of $Nf1^{fl/fl};Dmp1$-$Cre$ mice. To do this, we administered ponatinib intraperitoneally in 11-week-old $Nf1^{fl/fl};Dmp1$-$Cre$ mice for 5 weeks. Similar to the effect of additional MEKK2 deficiency on the $Nf1^{fl/fl};Dmp1$-$Cre$ phenotype, ponatinib-treated $Nf1^{fl/fl};Dmp1$-$Cre$ mice displayed a clear rescue of defects in cortical bone architecture (Fig. 4a, b), trabecular osteopenia (Fig. 4c, d), and calvarial hypomineralization (Fig. 4e and Supplementary Fig. 2) as compared to the vehicle group. Similarly, ponatinib treatment elevated P1NP, 25(OH) vitamin D, and phosphate levels relative to the levels in vehicle-treated $Nf1^{fl/fl};Dmp1$-$Cre$ mice (Fig. 4f). Similar to observations

made in vitro, $Nf1^{fl/fl};Dmp1$-$Cre$ mice following administration of ponatinib displayed significantly lower levels of p-MEKK2, p-MEK1, and p-ERK1/2 in osteoblasts compared to vehicle controls (Fig. 4g), but p-AKT levels were not changed (Supplementary Fig. 3a). Osteocyte dendritic network morphology was examined, but there was no clear evidence that ponatinib rescued this aspect of the NF1 loss-of-function phenotype (Supplementary Fig. 3b). As it may be clinically useful to target NF1-related skeletal therapies to discrete temporal windows, such as times of skeletal crisis, including fracture nonunion, or to critical windows in skeletal development, the durability of ponatinib effects were examined. Ponatinib treatment was initiated at 4 weeks of age, continued for 5 weeks, and then a phenotypes were assessed 7 weeks after drug withdrawal. Ponatinib treatment had durable effects that persist after treatment withdrawal under these conditions, including effects to ameliorate cortical defects, though, as expected, the magnitude of these effects was weaker than that seen with ponatinib treatment continuing until the experimental endpoint (Supplementary Fig. 4a–c). Taken together, these findings suggest that MEKK2 is a key MAP3K-mediating ERK activation in osteoblasts and that MEKK2 is a potential target for NF1 treatment.

**Discussion**

We here demonstrate rescue of the $Nf1^{fl/fl};Dmp1$-$Cre$ phenotype through genetic or pharmacologic inhibition of MEKK2. NF1 deficiency induces MEKK2 activation along with abnormal ERK activation, and MEKK2 contributes to the aberrant ERK activation seen with NF1 loss-of-function via the ability of MEKK2 to activate MEK1/2. MEKK2 also contributes to basal ERK activation, indicating that pathways other than those mediated by canonical RAF isoforms play key roles in ERK activation in osteoblasts. Despite $Mekk2^{-/-}$ mice displaying a low bone mass phenotype, additional MEKK2 deficiency in osteoblast-specific NF1-deficient mice dramatically improves NF1 skeletal phenotypes, including cortical bone porosity, trabecular bone mass, and endocrine disruptions related to calcium and phosphate homeostasis. Finding evidence that two loss-of-function alleles, each with strong deleterious phenotypes individually result in a phenotype markedly less severe than either alone when crossed provides clear evidence of epistasis, and therefore support for the biochemical model that MEKK2 here acts as a downstream mediator of the effects of NF1 deficiency in osteoblasts. We also note that, while there are several models of skeletal neurofibromatosis, each generated using different cre lines and with different features, several of the MEKK2 or ponatinib-dependent endpoints examined here, such as the cortical porosity/endocortical pitting phenotype, are shared across several models, including $Nf1^{fl/fl};Prx$-$Cre$[26], $Nf1^{fl/fl};Col2$-$Cre$[27],

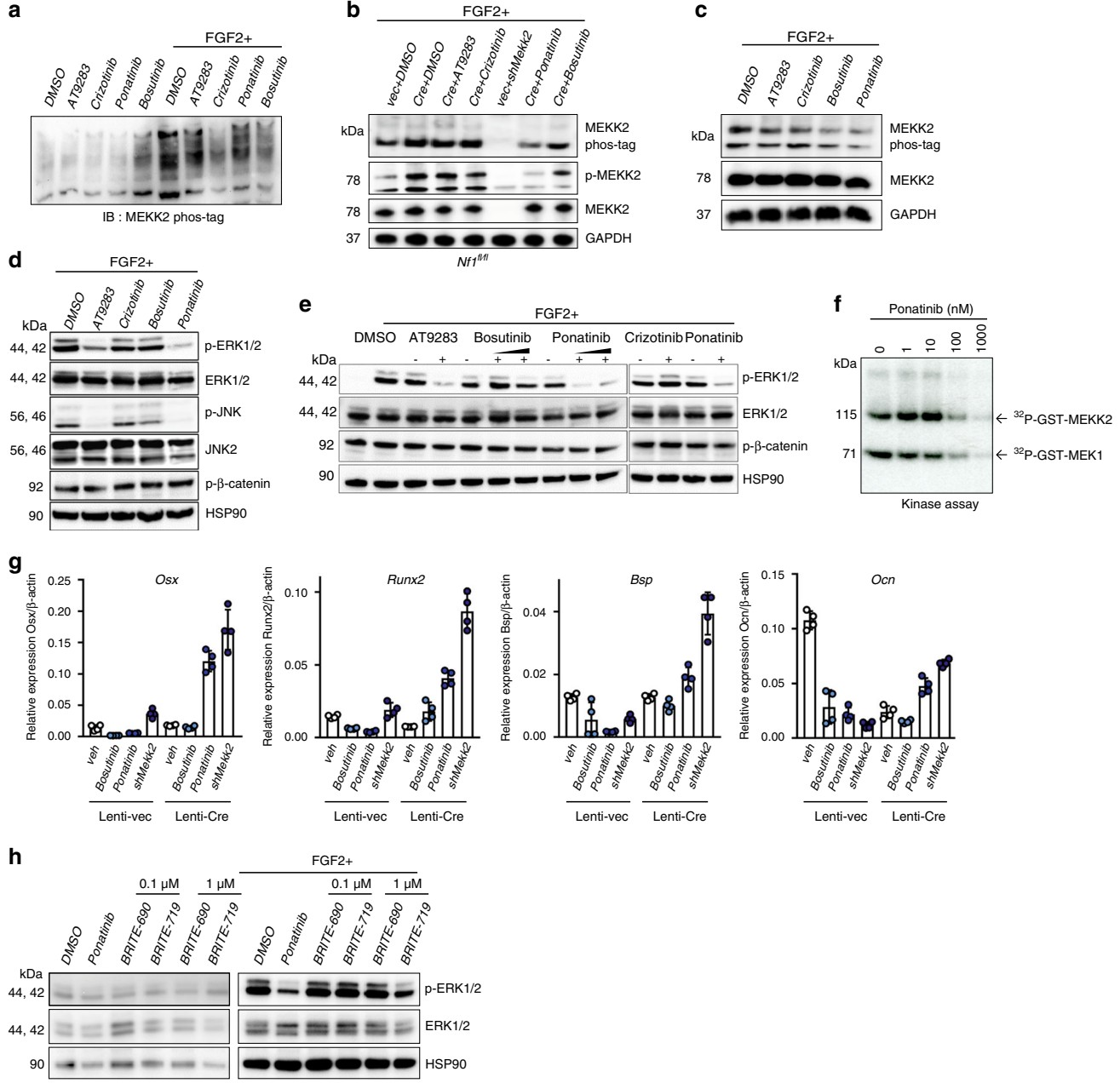

**Fig. 3 Pharmacologic inhibition of MEKK2 prohibits ERK activation. a** Primary COBs were immortalized through infection with a retrovirus expressing SV40 large T-Antigen. COBs were pre-treated with the indicated inhibitors (1 μM, 30 min), then stimulated with or without FGF2 (25 ng/ml, 20 min) 30 min later. MEKK2 phosphorylation was analyzed by phos-tag electrophoresis. **b** MEKK2 phosphorylation was analyzed by either phos-tag electrophoresis or immunoblotting with anti-p-MEKK2 in primary $Nf1^{fl/fl}$ osteoblasts infected with either vec or Cre lentivirus with the indicated inhibitors and FGF2 stimulation. **c** hMSCs were treated with the indicated inhibitors, then MEKK2 phosphorylation was examined by phos-tag electrophoresis. **d** WT immortalized COBs were lysed and immunoblotted for the indicated antibodies after inhibitor treatment. **e** Activation of ERK was detected in Saos-2 cells after incubation with the indicated inhibitors and stimulation with or without FGF2. **f** Purified unactivated GST-MEK1 was incubated with purified GST-MEKK2 and the indicated doses of ponatinib, and kinase activity of MEKK2 was analyzed by an in vitro kinase assay. **g** Expression levels of osteoblast genes in primary $Nf1^{fl/fl}$ osteoblasts infected with either vec or Cre lentivirus. After infection, these cells are treated with the indicated inhibitors or a *Mekk2*-targeting shRNA and cultured for 14 days ($n = 4$ biologically independent samples). mean ± s.d. **h** Saos-2 cells were treated with DMSO, ponatinib (1 μM), BRITE-0600690 (BRITE-690, inactive compound), and BRITE-0600719 (BRITE-719, active compound) for 1 h with or without FGF2 stimulation. Phosphorylation levels of ERK1/2 were assessed by immunoblotting. All data shown are representative of either two or three total independent experiments. All unprocessed blots are provided in Supplementary Fig. 5. Source data are provided as a Source Data file.

and $Nf1^{fl/fl};Osx-Cre$[32] mice, providing support for these findings having relevance beyond the specific *Dmp1-Cre* based model utilized here.

We here provide proof-of-concept that ponatinib (previously known as AP24534) can ameliorate skeletal pathology associated with NF1. In addition to the data offered in this study, support for the proposed mechanism of ponatinib mediating its effects through ERK pathway inhibition comes from studies finding that MEK1/2 inhibition promoted bone healing in the context of NF1 deficiency[2,33]. Many NF1-associated skeletal phenotypes, except

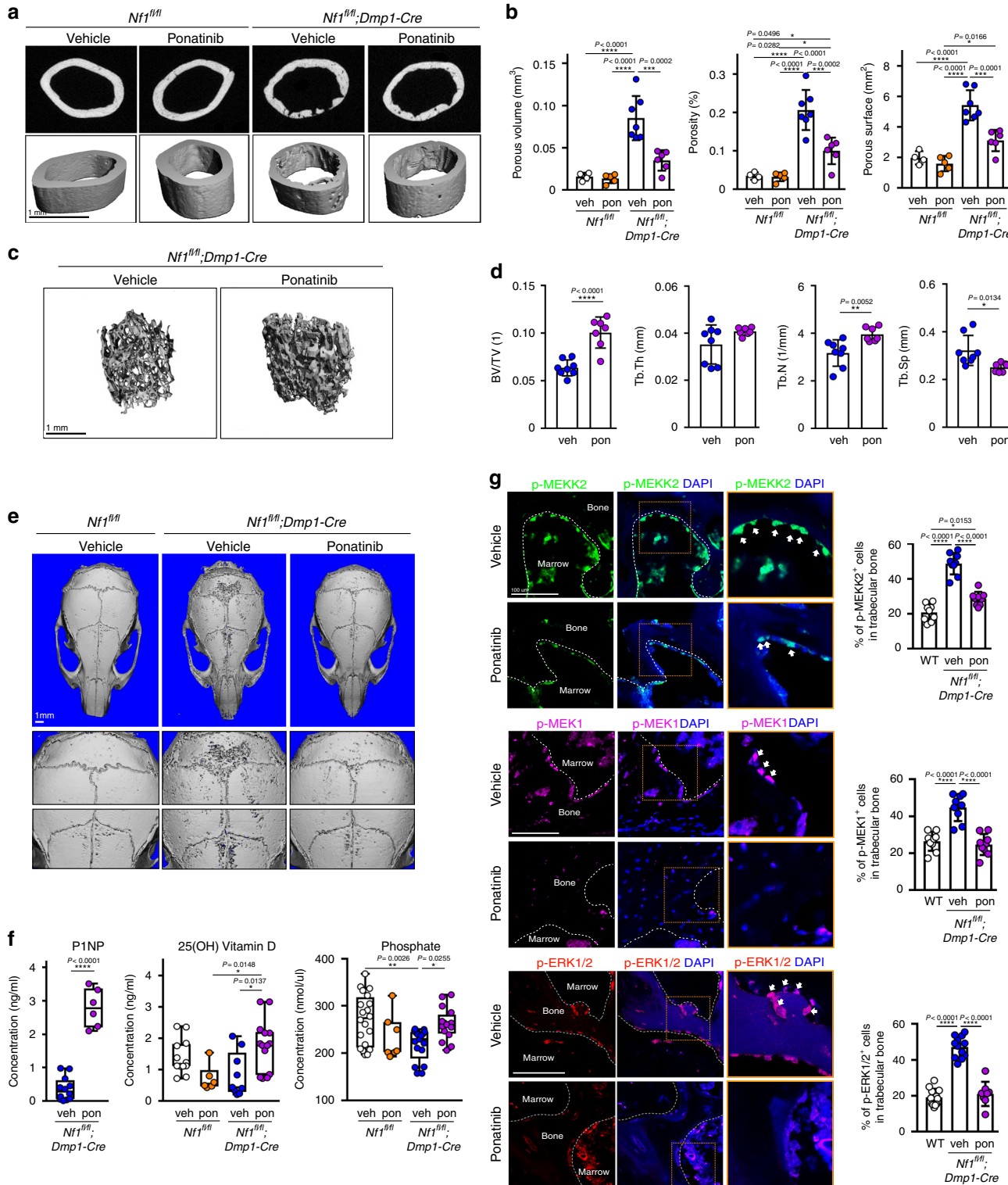

alterations in CTX levels, are rescued by ponatinib or MEKK2 deficiency. CTX levels not being regulated by MEKK2 is consistent with observations that, despite showing robust skeletal phenotypes, ablation of the ERK pathway in osteoblast lineage cells does not significantly alter CTX levels[9]. Further work has reported that MEK1/2 inhibition suppressed abnormal expression of genes regulating PPi homeostasis in NF1-deficient osteoblasts[32]. Thus, the findings here are broadly consistent with the role of the ERK pathway in the skeletal pathogenesis of NF1.

Ponatinib is currently approved by the FDA for patients with Philadelphia chromosome positive (Ph+) chronic myeloid leukemia (CML), particularly patients with tumors bearing the BCR-ABL T315I resistance mutation, suggesting that currently approved drugs may have an activity to ameliorate skeletal manifestations of NF1[34]. The clinical feasibility of using a MEKK2 inhibitor, either ponatinib or an alternative with increased specificity for MEKK2, is supported by observing that early treatment of mice produces a relatively durable amelioration of the phenotype that is maintained for several weeks after

**Fig. 4 Inhibition of MEKK2 ameliorates skeletal defects in a mouse model of skeletal NF1. a** Femurs from 16-week-old $Nf1^{fl/fl}$ mice treated with vehicle ($n = 4$) or ponatinib ($n = 5$) and $Nf1^{fl/fl};Dmp1$-$Cre$ mice treated with vehicle ($n = 7$) or ponatinib ($n = 6$) were analyzed by µCT. **b** Quantitative parameters are porous volume, porosity (porous volume/total volume), and porous surface from vehicle (veh) and ponatinib-treated groups (pon) of $Nf1^{fl/fl};Dmp1$-$Cre$ mice. Mean ± s.d., one-way ANOVA with Tukey's multiple comparison test. **c** Representative µCT 3D-reconstruction images of trabecular bone in the distal femur metaphysis. **d** Relative quantitative analysis of trabecular BV/TV, thickness (Tb.Th), trabecular number (Tb.N), and spacing (Tb.sp) in vehicle ($n = 8$) and ponatinib-treated ($n = 7$) group of $Nf1^{fl/fl};Dmp1$-$Cre$. Mean ± s.d., unpaired, two-tailed Student's $t$ test. **e** µCT scans of mouse skulls in 16-week-old $Nf1^{fl/fl}$ and $Nf1^{fl/fl};Dmp1$-$Cre$ mice treated with vehicle or ponatinib. **f** Serum levels of PINP, 25(OH) vitamin D, and phosphate in 16-week-old $Nf1^{fl/fl}$ and $Nf1^{fl/fl};Dmp1$-$Cre$ mice treated with vehicle (veh) or ponatinib (pon). Data are represented as box plots with the middle line representing the median, the box representing the 95% confidence interval of the median, and the whiskers representing the range. Each dot represents a separate mouse. **g** Representative immunofluorescent images and quantification for p-MEKK2 (green), p-MEK1 (magenta), and p-ERK (red) in femurs from 16-week-old $Nf1^{fl/fl}$ (WT) and $Nf1^{fl/fl};Dmp1$-$Cre$ mice treated with vehicle or ponatinib. Far right images show enlarged views of the dotted orange boxes. White arrows indicate signal positive osteoblasts. Nuclei are counterstained with DAPI (blue) and the scale bar indicates 100 µm. Three independent fields were examined per mouse ($n = 3$ mice per group). Mean ± s.d., one-way ANOVA with Tukey's multiple comparison test. *$P < 0.05$; **$P < 0.01$; ***$P < 0.001$; ****$P < 0.0001$. Source data are provided as a Source Data file.

treatment discontinuation. Chemically, the activity of ponatinib against MEKK2 observed here is overall consistent with observations that ponatinib displays relatively broad tyrosine-kinase activity, including against FGFR, VEGFR, PDGFR, FLT3, and c-Src, and it is possible or even probable that inhibition of non-MEKK2 targets, or other unexpected mechanisms of ponatinib, could have important contributions to the overall skeletal effects of ponatinib observed here[34–36]. However, support for MEKK2 being at least one relevant target of ponatinib comes from finding a very similar rescue of the $Nf1^{fl/fl};Dmp1$-$Cre$ phenotype with either ponatinib treatment or additional MEKK2 deficiency. Additional support comes from observing that a small molecule inhibitor of MEKK2 with a distinct chemical scaffold from ponatinib, but not an analog of this compound lacking MEKK2 activity, recapitulates some of the in vitro effects of ponatinib on ERK activation downstream of NF1 loss-of-function[31].

However, any therapeutic promise of ponatinib must be balanced against its toxicities, particularly concerns that ponatinib is associated with increased risk of arterial and venous thrombosis in CML patients[37]. Given the broad tyrosine-kinase activity of ponatinib, it is plausible or even likely that these toxicities may be associated with ponatinib targets other than MEKK2, suggesting that generation of analogs or novel chemotypes with improved relative selectivity for MEKK2 over other tyrosine kinases may allow the potential therapeutic effect of ponatinib to be chemically dissected away from limiting toxicities. Alternatively, strategies that focus on only providing transient therapy to resolve the most severe clinical sequelae, such as persistent nonunion after fracture, may offer a means to favorably balance the benefits and toxicities of these agents. Support for this strategy comes from studies that MEK1/2 inhibition can promote healing after fracture in models of NF1[2,20]. Additional support comes from observing that the effects of ponatinib persist for several weeks after therapy discontinuation, raising the possibility that either intermittent or crisis-targeted MEKK2 inhibition could have relatively durable overall effects on NF1-associated skeletal phenotypes extending beyond the period of treatment. Any consideration of the toxicities of either ponatinib or MEKK2 inhibition should be considered relative to alternative strategies, such as the broad ERK pathway inhibition associated with MEK inhibitors such as trametinib, which is potentially associated with cardiomyopathy, hepatic toxicity, hypertension, rash, and interstitial lung disease[38]. Both further preclinical and clinical studies will be needed to explore these issues and determine the potential utility of either ponatinib or alternative kinases inhibitors with increased MEKK2 selectivity. Additionally, as NF1 is a multi-system disorder, any novel therapeutic agent would have to be integrated into a comprehensive strategy for management of these complex patients.

We here identify MEKK2 as a mediator of aberrant ERK activation in NF1-deficient mature osteoblasts. To the degree that ERK is also implicated in the pathogenesis of other non-skeletal manifestations of NF1, including benign neurofibromas and malignant peripheral nerve sheath tumors[39], this suggests that evaluating the relative contributions of RAF isoforms versus MEKK2 in these non-skeletal contexts and potential therapeutic relevance of MEKK2 inhibition is warranted. Conversely, while there is evidence that ERK is a key mediator of skeletal and non-skeletal manifestations of NF1, several other pathways are activated downstream of NF1 loss-of-function, including PKA[40,41]. It will be of interest to determine whether MEKK2 contributes to the activation of these non-ERK pathways downstream of NF1 and also determine how inhibition of these alternative pathways may contribute to NF1 therapy. MEKK2-independent pathways acting downstream of NF1 are likely responsible for the few aspects of the $Nf1^{fl/fl};Dmp1$-$Cre$ phenotype not rescued by MEKK2 or ponatinib, including alterations in CTX levels or disruption of the osteocyte dendritic network. Given that MEKK2 can directly phosphorylate β-catenin and modulate its stability in osteoblasts and that aberrant β-catenin activation has been proposed as contributing to NF1-associated skeletal pathology[22,42], this raises the possibility that, in addition to ERK, β-catenin may also contribute to the therapeutic effects downstream of MEKK2 deficiency or inhibition on NF1.

## Methods

**Animals.** Floxed $Nf1$ ($Nf1^{fl/fl}$) mice (Stock 017639) were purchased from Jackson Laboratories. Generation of the $Mekk2^{-/-}$ mouse strain was described previously[43]. Transgenic mice expressing Cre recombinase under the control of the Dmp1 promoter ($Dmp1$-$Cre$)[44] were bred with $Nf1^{fl/fl}$ and/or $Mekk2^{-/-}$ mice. All mice used were backcrossed more than six generations onto the C57BL/6 background. All mice were maintained on a C57Bl6/J background throughout the study. All animals were maintained a 12 hr light/dark cycle, temperatures of 64–79 °F (~18–26 °C) with 40–60% humidity in accordance with the NIH Guide for the Care and Use of Laboratory Animals and were handled according to protocols approved by the Weill Cornell Medical College subcommittee on animal care (IACUC).

**Chemical reagents.** Kinase inhibitors used were AT9283 (cat. no. S1134, Selleckchem), crizotinib (PF-02341066) (cat. no. S1068, Selleckchem), ponatinib (AP24534) (cat. no. S1490, Selleckchem), bosutinib (SKI-606) (cat. no. S1014, Selleckchem), BRITE-0600690 (BRITE-690, inactive compound), and BRITE-0600719 (BRITE-719, compound 1s, active compound)[31].

**Osteoblast culture and differentiation assays.** Primary COBs were isolated from 5-day-old neonates by collagenase type II (cat. no. C6885, SigmaMillipore) and dispase II (cat. no. 04942078001, Roche) digestion. Cells were cultured in differentiation medium containing ascorbic acid and β-glycerophosphate. hMSCs were cultured and differentiated into osteoblasts as described by the manufacturer (Lonza). Primary COBs and Saos-2 cells were cultured in α-MEM medium (cat. no. 15-012CV, Cellgro) containing 10% FBS, 2 mM L-glutamine, 1% penicillin/streptomycin, 1% HEPES, and 1% nonessential amino acids and differentiated with

ascorbic acid and β-glycerophosphate. All cells were routinely tested to be myco-plasma negative.

**μCT analysis**. μCT analysis was conducted on a Scanco Medical μCT 35 system at the Citigroup Biomedical Imaging Core using the previously described[15]. Briefly, a Scanco Medical μCT 35 system with an isotropic voxel size of 7 μm was used to image the distal femur. Scans were conducted in 70% ethanol and used an X-ray tube potential of 55 kVp, an X-ray intensity of 0.145 mA, and an integration time of 600 ms. μCT analysis was performed by an investigator blinded to the genotypes of the animals under analysis. All endpoint μCT analysis was carried out on 16-week-old mice.

**In vitro kinase assay**. In all, 200 ng of recombinant GST-tagged MEKK2 (Dagger, cat. no. M0324, MilliporeSigma and Double dagger, cat. no. PV3822, Thermo-Fisher) was incubated for 20 min at 30 °C in kinase buffer (20 mM Hepes (pH 7.5), 20 mM MgCl2, 1 mM EDTA, 2 mM NaF,2 mM glycerophosphate, 1 mM DTT, and 10 μM ATP) and 10 μCi of (γ-32P) ATP (cat. no. NEG502H250UC, PerkinElmer) containing unactive GST-tagged MEK1 (cat. no. 14-420, MilliporeSigma), unactive His-tagged MEK2 (cat. no. 14-532, MilliporeSigma), or GST-tagged MEK5 (cat. no. A33380, ThermoFisher). The substrates then were resolved by SDS/PAGE, and phosphorylated proteins were visualized by autoradiography. To assess MEKK2 kinase activity after ponatinib treatment, recombinant proteins were incubated with serially diluted ponatinib for 20 min. Cell-free kinase assay conducted as described above.

**Immunoblotting**. Cells were lysed in RIPA lysis buffer (10 mM Tris, 50 mM NaCl, 5 mM EDTA, 2 mM NaF, 30 mM Peptin, 5 μg/ml Aprotinin, and 1% Triton X-100) supplemented protease and phosphatase inhibitor cocktail (cat. no. 78440, Ther-moFisher), and the protein concentration was determined by BCA protein assay reagent (cat. no. A53226, ThermoFisher). Proteins were separated by SDS-PAGE and then electrophoretically transferred to PVDF membranes. Membranes were incubated with a blocking buffer followed by incubation with primary and the appropriate secondary antibodies and developed with ECL blotting substrate (cat. no. 32134, ThermoFisher). Primary antibodies used were specific for NF1 (cat. no. A300-140A, BETHYL), MEKK2 (cat. no. A302-163A, BETHYL), ERK1/2 (cat. no. 9102, Cell Signaling Technology), phospho-ERK1/2 (Thr202/Tyr204) (cat. no. 4377, Cell Signaling Technology), AKT (cat. no. 9272, Cell Signaling Technology), phospho-AKT (Thr308) (cat. no. 13038, Cell Signaling Technology), phospho-p38 (Thr180/Tyr182) (cat. no. 9215, Cell Signaling Technology), phospho-RSK (Ser380) (cat. no. 12032, Cell Signaling Technology), phospho-GSK-3β (Ser9) (cat. no. 5558, Cell Signaling Technology), phospho-β-catenin (Ser675) (cat. no. 4176, Cell Signaling Technology), JNK2 (cat. no. 9258, Cell Signaling Technology), phospho-JNK (Thr183/Tyr185) (cat. no. 4668, Cell Signaling Technology), HSP90 (cat. no. sc-515081, Santa Cruz Biotechnology), and GAPDH (cat. no. sc-25778, Santa Cruz Biotechnology). Primary antibodies were detected with goat anti-mouse IgG HRP (cat no. 31430, Invitrogen) and goat anti-rabbit IgG HRP (cat no. 31460, Invitrogen). The generation of the phospho-MEKK2 polyclonal antibody was described previously[21]. To measure the levels of phosphorylated proteins, a Phos-tag gel (cat. no. 192-18001, Wako) was used according to the manufacturer's instructions. All uncropped blot images are provided in Supplementary Fig. 5.

**In vivo ponatinib treatment**. Treatment was initiated in 11-week-old male and female mice. Mice were administered ponatinib (25 mg/kg, intraperitoneal) (cat. no. S1490, Selleckchem) daily for 5 weeks. Ponatinib was first dissolved in DMSO at 100 mg/ml then resuspended in 30% PEG400 + 10% Triton X-100 in PBS. This solution was also used as the vehicle control. To determine the durability of the effects of ponatinib, treating mice starting at 4 weeks of age for 5 weeks and then subsequently waiting 7 weeks before evaluating the phenotypes.

**Serum assay**. We analyzed mouse-PINP (cat. no. LS-F7173, LSBio), mouse-CTX (cat. no. LS-F21349, LSBio), mouse-FGF23 (cat. no. LS-F31725-1, LSBio), mouse-PTH (cat.no. LS-F23085-1, LSBio), mouse-Phosphate (cat. no. ab65622, Abcam), and mouse-25(OH) Vitamin D (cat. no. ab213966, Abcam) by using a kit. All ELISAs were run according to the manufacturer's instructions.

**Immunostaining**. For immunofluorescence, femurs were harvested, embedded, and cut into frozen sections, followed by rehydration with PBS. Samples were then incubated with primary antibody against phospho-MEKK2 (Ser520; cat. no. PA5-105898, Invitrogen), phospho-MEK1 (Thr286; cat. no. 9127, Cell Signaling Tech-nology), phospho-ERK1/2 (Thr202/Tyr204; cat. no. 4370, Cell Signaling Tech-nology), phospho-AKT (Ser473; cat. no. 4060, Cell Signaling Technology) overnight at 4 °C, then washed three times with PBS. Secondary antibodies, goat anti-rabbit IgG Alexa fluor 633 (cat no. A-21070, Invitrogen) were added to the sample for 1 h followed by washing three times with PBS. Samples were finally mounted with ProLong Gold antifade reagent with DAPI (cat. no. P36931, Invi-trogen). For F-actin staining, fluorescent-conjugated phalloidin was used (cat. no. A12379, Invitrogen). Imaging was performed with a Zeiss LSM 880 with Airyscan high-resolution-detector confocal microscope. All immunofluorescence

experiments were quantified by examining at least three independent fields per mouse (n = 3 mice per group). For histological analysis, femurs were dissected from mice, fixed in 4% PFA for 24 h, and decalcified by daily changes of 15% tetrasodium EDTA for 2 weeks and embedded and cut into frozen sections, fol-lowed by rehydration with PBS. Decalcified sections were incubated with primary antibodies to phospho-MEKK2 (Ser520) (cat. no. PA5-105898, Invitrogen) over-night at 4 °C and then immunoreactivity was visualized using HRP substrate (cat. no. SK-4805, Vector lab). Methyl green was used as counterstain.

**qRT-PCR**. Total mRNA was isolated from calvarial osteoblasts using TRIzol (cat. no. 15596026, Invitrogen), and reverse transcription was performed with the High-Capacity cDNA Reverse Transcription Kit (cat. no. 4368814, Applied Biosystems) according to the manufacturer's instructions. The primer sequences used in this study are described in Supplementary Table 1.

**Statistical analyses**. All data are shown as the mean ± s.d. or mean ± s.e.m. as indicated. For comparisons between two groups unpaired, two-tailed Student's $t$ tests were used. For comparisons of three or more groups, one-way ANOVA was used if normality tests passed, followed by Tukey's multiple comparison test for all pairs of groups. GraphPad PRISM v.7.04 was used for statistical analysis. $P < 0.05$ was considered statistically significant. $*P < 0.05$, $**P < 0.01$, $***P < 0.001$, and $****P < 0.0001$.

**Reporting summary**. Further information on research design is available in the Nature Research Reporting Summary linked to this article.

## Data availability
All relevant data are available from the authors. Source data are provided with this paper.

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

## Acknowledgements

This project was supported by a Career Award for Medical Scientists from the Burroughs Wellcome Fund, the NIH under award DP5OD021351 and R01AR075585, the Department of Defense under NF150055. This publication is based on research supported by the Pershing Square Sohn Cancer Research Alliance via an award to M.B.G. This research was supported by the award to D.Y.S. through the Basic Science Research Program of the National Research Foundation of Korea (NRF) funded by the Ministry of Education (Grant no. 2017R1A6A3A03003075). We thank Graham Su at Yale University, the Research Animal Resource Center (RARC), and the Microscopy and Image Analysis Core Facilities at Weill Cornell Medicine for their technical assistance.

## Author contributions

M.B.G. and J.H.S. supervised these studies and generated the project concept. M.B.G. and S.B. drafted the manuscript. S.B., D.Y.S., A.R.Y., M.E., M.C., R.X., N.L., and J.S. conducted experiments. B.S., J.E.S., and A.L.W. provided critical support and feedback for in vivo studies.

## Competing interests

The authors declare no competing interests.
