## [Peer Review File · Nature Communications]

Reviewers' Comments:

Reviewer #1:

Remarks to the Author:

The manuscript by Bok and colleagues reveals that MEKK2 mediates ERK activation signaling in the skeleton using mice with conditional loss of the neurofibromatosis-1 (Nf1) gene in osteoclasts. In their study, they use two strains of mice, each with skeletal defects, to demonstrate that Mekk2 loss rescues the skeletal defect in Nf1 conditional knockout mice.

What accounts for the MEKK2-independent, but neurofibromin-dependent, regulation of CTX? What other subtle Nf1 mutant skeletal abnormalities are not rescued by MEKK2 inhibition?

The p-MEKK2 block contains no control for total MEKK2 protein expression, and it is not clear which of the bands in the chain of bands is MEKK2 (Fig 3a). There are also no total protein loading controls for phospho-JNK (Fig 3b) and phospho-ERK1/2 (Fig 3c).

The data presented in Figure 4 suggests that cortical bone density is completely correctable in adult mice, even though this defect is manifest at much earlier ages. Are the other Nf1-mutant skeletal defects observed in these mice similarly rescued with ponatinib at this time point?

Given the potential translational importance of these findings, have the authors examined other strains of Nf1 mutant mice with skeletal defects, and/or established treatment windows for durable effects. As the authors appreciate, the pseudarthrosis in children can be devastating, and treatments would need to be administered during early infancy/childhood.

Reviewer #2:

Remarks to the Author:

This manuscript by Bok and coworkers proposes a novel approach for treating neurofibromatosis type 1 based on genetic or pharmacological inhibition of MEKK2. The authors provide evidence that MEKK2 is active in osteoblasts where it is necessary for activation of ERK1/2, RSK, GSK3 β and AKT. Furthermore, crossing Dmp1Cre;Nf1 f/f mice with Mekk2^{-/-} mice is shown to partially rescue the osteopenic phenotype of Dmp1Cre;Nf1 f/f mice, a model for neurofibromatosis type 1. Similarly, the broad-spectrum tyrosine kinase inhibitor, ponatinib, which inhibits MEKK2, is also shown to partially rescue the cortical bone phenotype of Dmp1Cre;Nf1 f/f mice. These are potentially very important findings with high potential for clinical translation. However, a number of issues need to be addressed to strengthen the overall impact of this work.

1. In fig 1, MEKK2 deficiency is shown to block known NF1-related signaling events in cell culture including RSK, ERK, GSK3 β , and AKT. Furthermore, lentiCre-mediated knockout of Nf1 is shown to activate ERK and AKT in Nf1 f/f COB cells. However, effects of Nf1 knockout on MEKK2 activity are not shown even though this is central to the main hypothesis of the manuscript. The ShRNA Nf1 knockdown in MSCs provides only weak evidence for increased activation of MEKK2 in the absence of FGF stimulation (Fig1 ef). Does Nf1 knockout consistently increase MEKK2 activity? Also no information about reproducibility or statistical analysis is provided for any of the studies shown in Fig 1.

2. Fig 2 shows that crossing Dmp1Cre;Nf1 f/f mice with Mekk2^{-/-} mice is able to partially reverse the increase in cortical porosity as well as increase the bone formation marker, PINP. Immunofluorescent data is also included suggesting that bone-associated pERK is increased in Dmp1Cre;Nf1 f/f mice and that this is reversed by the Mekk2^{-/-} cross. However, additional characterization of resulting bone phenotypes is needed. What are effects of Nf1 knockout +/- Mekk2^{-/-} on trabecular bone since this is also affected in neurofibromatosis and Dmp1Cre;Nf1 f/f mice? Also, Dmp1Cre;Nf1 f/f mice have major endocrine disruptions including

elevated FGF23 and PTH , altered mineral homeostasis and abnormal osteocyte structure. Are these also rescued by Mekk2 disruption? Finally, pERK results need to be quantified with statistical analysis. The significance of this result could be further strengthened if pMEKK2 were measured in vivo together with related signaling intermediates such as pRSK, pGSK3 β , and pAKT.

3. Fig 3 screens several tyrosine kinase inhibitors for ability to inhibit MEKK2 signaling. Panel A does not provide convincing data that ponatinib significantly reduces pMEKK2. The ability of this compound to inhibit other downstream signals is consistent with its broad effects on tyrosine kinases and does not prove that its actions are mainly explained by MEKK2 inhibition.

4. In vivo analysis of ponatinib in Dmp1Cre;Nf1 f/f mice are subject to the same criticisms raised for the data in fig 2 (i.e. does it reverse all abnormalities in Dmp1Cre;Nf1 f/f mice?, quantitation of IF results, examination of other signaling markers). From this data, it is not clear that effects of ponatinib are necessarily related to its inhibition of MEKK2.

Reviewer #3:

None

Point-by-point response

(Reviewer comments are in red, our response is in black)

Reviewers' comments:

Reviewer #1 (Remarks to the Author):

The manuscript by Bok and colleagues reveals that MEKK2 mediates ERK activation signaling in the skeleton using mice with conditional loss of the neurofibromatosis-1 (Nf1) gene in osteoclasts. In their study, they use two strains of mice, each with skeletal defects, to demonstrate that *Mekk2* loss rescues the skeletal defect in Nf1 conditional knockout mice.

What accounts for the MEKK2-independent, but neurofibromin-dependent, regulation of CTX? What other subtle Nf1 mutant skeletal abnormalities are not rescued by MEKK2 inhibition?

We share the reviewer's interest in this question. First, we note that we previously did not find a difference in serum CTX in mice with conditional inactivation of the ERK pathway in osteoblasts (Kim et al. Int J Mol Sci. 2019). Thus, the finding that CTX levels appear to be MEKK2-independent in this setting is overall consistent with the proposed role for MEKK2 in the ERK pathway in this context. During revisions we have performed additional studies to identify NF1-dependent MEKK2-independent pathways, including, performing immunofluorescence for pAKT levels. pAKT does not appear to be regulated by either MEKK2 or ponatinib treatment and did not display clear differences between *Nf1*^{dmp1} or control groups (added as Supplementary Figure 1a and 3a). Thus, pAKT does not appear to be the NF1-dependent, MEKK2-independent pathway responsible for this effect. Ultimately, it is unclear which pathways downstream of NF1 loss-of-function are MEKK2-independent and responsible for the phenotypes such as changes in CTX. We propose that this issue deserves further separate subsequent further evaluation especially due to the difficulty in modeling the effects on CTX levels due to physiologic osteoblast-osteoclast crosstalk in vitro. A discussion of possible NF1-dependent pathways that may contribute to the phenotypes here in an MEKK2 independent manner has been added to the manuscript.

To address the question of which phenotypes may be NF1 dependent but MEKK2 independent, during revisions we have performed extensive additional phenotyping of *Nf1*^{dmp1}, *Mekk2*^{-/-}, *Nf1*^{dmp1} *Mekk2*^{-/-}, and ponatinib-treated *Nf1*^{dmp1} mice. However, nearly all of this additional data, including analysis of trabecular bone mass, calvarial phenotypes, P1NP, 25OH Vitamin D, phosphate, FGF23 levels, and expression of osteoblast marker genes during in vitro differentiation find phenotypes that are dependent on NF1 and ameliorated or reversed by additional MEKK2 deficiency in line with our major hypothesis. However, potential exceptions were observed. Most notably, as previously reported Nf1-deficiency causes disruption of the osteocyte dendritic network (Kamiya et al. JBMR 2017), and it does not appear as though this aspect of the phenotype is fully restored by either additional MEKK2 deficiency or ponatinib treatment (Supplementary Figure 1b and 3b).

The p-MEKK2 block contains no control for total MEKK2 protein expression, and it is not clear which of the bands in the chain of bands is MEKK2 (Fig 3a). There are also no total protein loading controls for phospho-JNK (Fig 3b) and phospho-ERK1/2 (Fig 3c).

Knockdown of MEKK2 was performed to clarify which band represents MEKK2 (added as new Figure 3b). During revisions, we have repeated these immunoblotting experiments to overall increase the quality of these images and also add additional loading controls (added as new Figure 3d and 3e).

The data presented in Figure 4 suggests that cortical bone density is completely correctable in adult mice, even though this defect is manifest at much earlier ages. Are the other *Nf1*-mutant skeletal defects observed in these mice similarly rescued with ponatinib at this time point?

We thank the reviewer for this comment, as we believe that addressing this has substantially improved the revised manuscript. During revisions, we have added additional measurement of trabecular bone mass and calvarial mineralization, finding that these also show clear partial or even full rescue with both additional MEKK2 deficiency or ponatinib treatment (added as new Figure 2c-e, 4c-e, and Supplementary Figure 2). We note that the rescue of trabecular bone mass in *Nf1^{dmp1}Mekk2^{-/-}* mice occurs despite both *Nf1^{dmp1}* and *Mekk2^{-/-}* mice each separately displaying moderate to severe trabecular osteopenia. Additionally, ponatinib treatment or additional MEKK2 deficiency improved other parameters, including serum levels of phosphate, the anabolic bone turnover marker P1NP, or levels of selected osteoblast markers such as osteocalcin (Ocn) during in vitro differentiation assays (added as new Figure 2f, 3g, and 4f). Thus, in the revised dataset, we find that a very broad range of NF1 loss-of-function associated phenotypes are rescued by either additional MEKK2 deficiency or ponatinib treatment.

Given the potential translational importance of these findings, have the authors examined other strains of *Nf1* mutant mice with skeletal defects, and/or established treatment windows for durable effects. As the authors appreciate, the pseudarthrosis in children can be devastating, and treatments would need to be administered during early infancy/childhood.

While it was not feasible to breed/obtain and test additional NF1-deficient mouse strains within the time limit of revisions in addition to COVID-19 related restrictions on mouse breeding and import, we have conducted a study to determine the durability of the effects of ponatinib, treating *Nf1^{dmp1}* mice starting at 4 weeks of age for 5 weeks and then subsequently waiting 7 weeks before evaluating the phenotype to observe the durability of these effects. Indeed, in this study a durable effect of ponatinib treatment is seen, though, as expected, the magnitude of these effects is weaker than that seen with ponatinib treatment continuing until the experimental endpoint (added as Supplementary Figure 4a-c). We believe that this raises data raises the feasibility of MEKK2 inhibition as a potential strategy to treat neurofibromatosis by suggesting that treatment during key windows during skeletal development may have relatively durable effects, providing a strategy to maximize clinical benefit while minimizing toxicities. Further studies will be needed to determine how MEKK2 inhibitors would best fit into a comprehensive strategy to manage the skeletal effects of neurofibromatosis. Discussion of these points has been enhanced in the revised manuscript.

Reviewer #2 (Remarks to the Author):

This manuscript by Bok and coworkers proposes a novel approach for treating neurofibromatosis type 1 based on genetic or pharmacological inhibition of MEKK2. The authors provide evidence that MEKK2 is active in osteoblasts where it is necessary for activation of ERK1/2, RSK, GSK3 β and AKT. Furthermore, crossing Dmp1Cre;Nf1 f/f mice with Mekk2 $^{-/-}$ mice is shown to partially rescue the osteopenic phenotype of Dmp1Cre;Nf1 f/f mice, a model for neurofibromatosis type 1. Similarly, the broad-spectrum tyrosine kinase inhibitor, ponatinib, which inhibits MEKK2, is also shown to partially rescue the cortical bone phenotype of Dmp1Cre;Nf1 f/f mice. These are potentially very important findings with high potential for clinical translation. However, a number of issues need to be addressed to strengthen the overall impact of this work.

1. In fig 1, MEKK2 deficiency is shown to block known NF1-related signaling events in cell culture including RSK, ERK, GSK3 β , and AKT. Furthermore, lentiCre-mediated knockout of Nf1 is shown to activate ERK and AKT in Nf1 f/f COB cells. However, effects of Nf1 knockout on MEKK2 activity are not shown even though this is central to the main hypothesis of the manuscript. The ShRNA Nf1 knockdown in MSCs provides only weak evidence for increased activation of MEKK2 in the absence of FGF stimulation (Fig1 ef). Does Nf1 knockout consistently increase MEKK2 activity? Also no information about reproducibility or statistical analysis is provided for any of the studies shown in Fig 1.

During revisions, this aspect of the manuscript has been strengthened with additional data. In particular, we find that either shRNA mediated knockdown of NF1 or transduction of *Nf1*^{fl/fl} osteoblasts with cre lentivirus increases MEKK2 phosphorylation as shown by either use of a phos-tag gel or immunoblotting with anti-pMEKK2 (Zhang et al. EMBO J 2006) (Figure 1f and added as new Figure 1g). In this additional data we observe activation of MEKK2 under both basal conditions and with FGF2 stimulation. Additionally during revisions, we have performed immunohistochemistry for pMEKK2 levels in vivo, finding that *Nf1*^{dmp1} mice display greater numbers of pMEKK2+ skeletal mesenchymal cells (added as new Figure 1h). Thus, the revised manuscript now has stronger support for the contention that *Nf1* loss-of-function activates a MEKK2-MEK-ERK pathway both in vivo and in vitro.

Regarding the reproducibility of experiments in Fig 1, information on replicates has been added to the figure legend. All experiments were performed for a minimum of 2 or 3 total independent repeats.

2. Fig 2 shows that crossing Dmp1Cre;Nf1 f/f mice with Mekk2 $^{-/-}$ mice is able to partially reverse the increase in cortical porosity as well as increase the bone formation marker, PINP. Immunofluorescent data is also included suggesting that bone-associated pERK is increased in Dmp1Cre;Nf1 f/f mice and that this is reversed by the Mekk2 $^{-/-}$ cross.

However, additional characterization of resulting bone phenotypes is needed. What are effects of Nf1 knockout +/- Mekk2 $^{-/-}$ on trabecular bone since this is also affected in neurofibromatosis and Dmp1Cre;Nf1 f/f mice? Also, Dmp1Cre;Nf1 f/f mice have major endocrine disruptions including elevated FGF23 and PTH, altered mineral homeostasis and abnormal osteocyte structure. Are these also rescued by Mekk2 disruption? Finally, pERK results need to be quantified with statistical analysis. The significance of this result could be further strengthened if pMEKK2 were measured in vivo together with related signaling intermediates such as pRSK, pGSK3 β , and pAKT.

We thank the reviewer for these suggestions (including the similar suggestion #4 below), as they provided a insightful guide to uncover additional MEKK2-dependent aspects of the NF1 loss-of-function phenotype and

thereby strengthen this study. As suggested, trabecular bone phenotypes were examined, finding that both *Nf1*^{dmp1} and *Mekk2*^{-/-} mice displayed osteopenia as reported (Kamiya et al. JBMR 2017 or Greenblatt et al. PNAS 2016) This osteopenia was rescued or even fully reversed either in dual knockout *Nf1*^{dmp1}*Mekk2*^{-/-} mice. Similarly, 25OH vitamin D levels, phosphate levels and FGF23 levels were all normalized in *Nf1*^{dmp1}*Mekk2*^{-/-} relative to *Nf1*^{dmp1} mice. P1NP levels were also increased in *Nf1*^{dmp1}*Mekk2*^{-/-} relative to *Nf1*^{dmp1} mice.

We also examined the osteocyte dendritic network as suggested. While we observe clear disruption of the osteocyte network in *Nf1*^{dmp1} mice, this aspect of the *Nf1* loss-of-function phenotype does not show obvious or a substantial degree of rescue by additional MEKK2 deficiency (added as Supplementary Figure 1b and 3b).

pAKT levels were also examined by immunofluorescence, but pAKT did not appear to be regulated by either MEKK2 or ponatinib treatment and did not display clear differences between *Nf1*^{dmp1} or control groups (added as Supplementary Fig. 1a and 3a). This finding is consistent with the apparent lack of effect on pAKT levels observed in NF1-deficient osteoblasts in prior reports such as Elefteriou et al. Cell Metabolism 2006.

As requested, immunofluorescence studies, including pERK, pMEK1, and pMEKK2, have been performed and quantitated with statistical analysis provided (added as new Figure 2g and 4g), finding that robust and statistically significant differences in staining are present, consistent with our prior qualitative interpretation.

3. Fig 3 screens several tyrosine kinase inhibitors for ability to inhibit MEKK2 signaling. Panel A does not provide convincing data that ponatinib significantly reduces pMEKK2. The ability of this compound to inhibit other downstream signals is consistent with its broad effects on tyrosine kinases and does not prove that its actions are mainly explained by MEKK2 inhibition.

This point is well taken, and during revisions we have generated additional data to strengthen the evidence that ponatinib acts through MEKK2. First, to address the concern that the data in the initial submission did not provide convincing biochemical evidence that ponatinib inhibits MEKK2, new experiments have been added. In Fig 3b and 3c, both mouse and human cells were used to determine that ponatinib blocks MEKK2 activation as shown by both pMEKK2 immunoblotting and a MEKK2 phos-tag assay. These data are matched by similar new in vivo immunofluorescence data showing similar decreases in pMEKK2, pMEK1 and pERK staining in osteoblasts after ponatinib treatment. Additionally, several of the other experiments in Figure 3 to establish the biochemical effects of ponatinib in osteoblasts were repeated and overall improved in quality through the addition of new data, including the revised Fig 3d and 3e. We have also added additional in vitro data to show that ponatinib phenocopies the effects of shMEKK2 knockdown on osteoblast marker gene expression during in vitro differentiation assays (added as new Figure 3g). We additionally note that ponatinib has relatively potent (IC50 ~100nM) activity to block MEK1 phosphorylation by MEKK2 in a cell free in vitro kinase assay, confirming that these effects are direct (Figure 3f). In addition to this data, we primarily base our conclusions on the in vivo studies comparing ponatinib treatment of *Nf1*^{dmp1} mice to the effects of additional MEKK2 deficiency, which have now been strengthened with additional data during revisions.

To additionally specifically strengthen the evidence that ponatinib is working through inhibition of MEKK2, during revisions we have obtained a small molecule expressly identified as an MEKK2 inhibitor with a completely distinct chemical scaffold than that of ponatinib (BRITE-719, Ahmad et al. BBRC 2018). As added in Figure 3h, ponatinib and BRITE-719 reduced pERK1/2 activation in response to FGF2, and an analogue of BRITE-719 that lacks activity against MEKK2 had no effect. Thus, an independent MEKK2 inhibitor can also

recapitulate the key in vitro finding that MEKK2 inhibition attenuates NF1 loss-of-function associated ERK pathway activation. Taking all of the above together, we believe that this offers an overall compelling package of data to support that ponatinib is acting, at least in part, by inhibiting MEKK2.

However, despite this additional data, we share the overall caution expressed regarding the specificity of ponatinib effect on MEKK2, as the literature provides many examples of new targets or alternative mechanisms of action being identified for drugs that were thought to be well understood after decades of intense study (Hawley et al. *Science* 2012 or Sievers et al. *Science* 2018). In light of this, we have included a discussion of the possibility that ponatinib only partially acts through MEKK2 and emphasize that ponatinib may also engage other targets that are important for the skeletal effects reported here.

4. In vivo analysis of ponatinib in *Dmp1Cre;Nf1 f/f* mice are subject to the same criticisms raised for the data in fig 2 (i.e. does it reverse all abnormalities in *Dmp1Cre;Nf1 f/f* mice?, quantitation of IF results, examination of other signaling markers). From this data, it is not clear that effects of ponatinib are necessarily related to its inhibition of MEKK2.

To address this issue, we have taken a similar approach as used to enhance the analysis of phenotypic rescue in *Nf1^{dmp1}Mekk2^{-/-}* mice above. As with *Nf1^{dmp1}Mekk2^{-/-}* mice, ponatinib-treated *Nf1^{dmp1}* mice were analyzed for trabecular bone mass and calvarial mineralization, and both parameters were rescued by ponatinib treatment (added as new Figure 4c-e). Similarly, ponatinib treatment elevated 25OH vitamin D levels and phosphate levels and substantially increased P1NP levels relative to vehicle treated *Nf1^{dmp1}* mice (added as new Figure 4f). Osteocyte dendritic network morphology was examined, but there was not clear evidence that ponatinib rescued this aspect of the NF1-loss-of-function phenotype (added as Supplementary Figure 3b). In terms of signaling markers, immunofluorescence studies showed that, as expected, ponatinib reduced levels of pMEKK2, pMEK1, and pERK1/2 in line with our proposed mechanism (added as new Figure 4g, left). Additionally, as requested, immunofluorescence studies have been quantitated with statistical analysis provided, demonstrating that the qualitative differences observed indeed translate into statistically significant quantitative differences (added as new Figure 4g, right). A similar ability of ponatinib to rescue or increase expression of osteoblast marker genes during in vitro differentiation assays, including *Osx*, *Runx2*, *Bsp* and *Ocn*, was also noted (Figure 3g).

Overall this additional data both strengthens the evidence that ponatinib provides a strong reversal of many aspects of the *Nf1^{dmp1}* phenotype, and when considered alongside the very similar data obtained during revisions on *Nf1^{dmp1}Mekk2^{-/-}* mice, that ponatinib is likely to work at least in part through MEKK2.

Reviewers' Comments:

Reviewer #1:

Remarks to the Author:

The authors have responded to all comments in a satisfactory manner. The only issue remaining is the mechanism, for which they performed a single molecule analysis. It would have been helpful to examine other downstream signaling pathways, like cAMP, which could potentially play MEKK-independent roles.

Reviewer #2:

Remarks to the Author:

This revised manuscript now makes a convincing case that at least a portion of the neurofibromatosis type 1 phenotype is explained by elevation in activity of MEKK2 and that genetic or pharmacological inhibition of this kinase can correct many of the features of this disorder. This study opens the possibility of using MEKK2 inhibitors to treat NF1, offering new potential options for patients.

Reviewer #3:

None

Point-by-point response

(Reviewer comments are in red, our response is in black)

REVIEWERS' COMMENTS:

Reviewer #1 (Remarks to the Author):

The authors have responded to all comments in a satisfactory manner. The only issue remaining is the mechanism, for which they performed a single molecule analysis. It would have been helpful to examine other downstream signaling pathways, like cAMP, which could potentially play MEKK-independent roles.

We thank the reviewer for the prior suggestions that we believe have resulted in an improved manuscript over the review process. We share the reviewer's interest in understanding which pathways, in addition to the ERK pathway, may be acting downstream of both NF1 and MEKK2. For this reason, we have assessed activation of the p38, AKT and PI3K pathways, and did not find evidence of regulation by MEKK2 (Fig. 1a, 1d, and Supplementary Figure 1a). We have begun assessment of the relationship of MEKK2 to cAMP generation downstream of NF1 loss-of-function, but find that these effects may be highly context dependent and vary based on details of the culture system utilized. Therefore, we believe that separate, dedicated investigation of this important topic will be needed to provide a high confidence answer to this question.

Reviewer #2 (Remarks to the Author):

This revised manuscript now makes a convincing case that at least a portion of the neurofibromatosis type 1 phenotype is explained by elevation in activity of MEKK2 and that genetic or pharmacological inhibition of this kinase can correct many of the features of this disorder. This study opens the possibility of using MEKK2 inhibitors to treat NF1, offering new potential options for patients.

We thank the reviewer for the constructive comments that improved this manuscript.